# A gene-expression screen identifies a non-toxic sumoylation inhibitor that mimics SUMO-less human LRH-1 in liver

Miyuki Suzawa[1†], Diego A Miranda[1†], Karmela A Ramos[1†], Kenny K-H Ang[2], Emily J Faivre[1], Christopher G Wilson[2], Laura Caboni[3], Michelle R Arkin[2], Yeong-Sang Kim[4], Robert J Fletterick[3], Aaron Diaz[5], John S Schneekloth Jr[4], Holly A Ingraham[1*]

[1]Department of Cellular and Molecular Pharmacology, University of California, San Francisco, San Francisco, United States; [2]Small Molecule Discovery Center, Department of Pharmaceutical Chemistry, University of California, San Francisco, San Francisco, United States; [3]Department of Biochemistry and Biophysics, University of California, San Francisco, San Francisco, United States; [4]Chemical Biology Laboratory, National Cancer Institute, Frederick, United States; [5]Department of Epidemiology and Biostatistics, University of California, San Francisco, San Francisco, United States

**Abstract** SUMO-modification of nuclear proteins has profound effects on gene expression. However, non-toxic chemical tools that modulate sumoylation in cells are lacking. Here, to identify small molecule sumoylation inhibitors we developed a cell-based screen that focused on the well-sumoylated substrate, human Liver Receptor Homolog-1 (hLRH-1, NR5A2). Our primary gene-expression screen assayed two SUMO-sensitive transcripts, *APOC3* and *MUC1*, that are upregulated by SUMO-less hLRH-1 or by siUBC9 knockdown, respectively. A polyphenol, tannic acid (TA) emerged as a potent sumoylation inhibitor in vitro (IC$_{50}$ = 12.8 μM) and in cells. TA also increased hLRH-1 occupancy on SUMO-sensitive transcripts. Most significantly, when tested in humanized mouse primary hepatocytes, TA inhibits hLRH-1 sumoylation and induces SUMO-sensitive genes, thereby recapitulating the effects of expressing SUMO-less hLRH-1 in mouse liver. Our findings underscore the benefits of phenotypic screening for targeting post-translational modifications, and illustrate the potential utility of TA for probing the cellular consequences of sumoylation.

*For correspondence: holly. ingraham@ucsf.edu

†These authors contributed equally to this work

Competing interests: The authors declares that no competing interests exist.

## Introduction

SUMO-modification or sumoylation with the Small Ubiquitin-like Modifier (SUMO) is a prevalent post-translational modification of many transcription factors and is generally associated with transcriptional repression (*Gill, 2005*). Similar to other ubiquitin-like modifications, the sumoylation cycle is multi-stepped, as reviewed in *Gareau and Lima (2010*), and is initiated by E1 (SAE1), which forms a thioester bond with either SUMO-1, 2, or 3. The single E2 (UBC9) facilitates the hand-off and covalent conjugation of SUMO on a given protein substrate. Although E3s are believed to guide substrate selectivity in cells, only E1 and E2 are required for in vitro sumoylation (IVS). Sumoylation is then reversed by sentrin-specific proteases (SENPs).

Genetic manipulations that either eliminate the sumoylation machinery or permanently disrupt the normal sumoylation cycle of a substrate can result in embryonic lethality or impaired organogenesis (*Flotho and Melchior, 2013*; *Kang et al., 2010*; *Lee et al., 2011a*; *Nacerddine et al., 2005*;

**eLife digest** Proteins in cells carry out diverse tasks. One way in which this diversity is achieved by proteins is through the attachment of molecular tags. SUMO is one such tag that can reversibly attach to proteins and alter their activity. The modification of proteins by SUMO is known as sumoylation, and it regulates many processes that are essential for living cells. In particular, transcription factors—the proteins that bind to DNA to switch genes on or off—are highly modified by SUMO. However, the consequences of sumoylation are not fully understood, and current research into this area has been hindered by a lack of effective and non-toxic chemicals that stop or slow down sumoylation.

Suzawa, Miranda, Ramos et al. have now screened a large collection of compounds, which had already been approved for medical use, to find one that could inhibit sumoylation without toxic effects. The compounds were tested for their ability to alter the activity of a transcription factor called human Liver Receptor Homolog-1. This protein, which is referred to as LRH-1 for short, is an ideal candidate to test SUMO inhibitors because it is highly modified by multiple SUMO tags.

This screen identified a compound from plants called tannic acid as a non-toxic and potent inhibitor of sumoylation. Further experiments confirmed that tannic acid prevented the modification of LHR-1 as well a number of different proteins that also commonly modified by SUMO. Inhibiting the sumoylation of LRH-1 led to an increase in the expression of genes that are normally silenced by SUMO-modified LRH-1. Similar results were obtained when tannic acid was tested using human cells and "humanized" liver cells from mice that had been engineered to express human LRH-1. The next big challenge is to find new chemical probes that can be used to specifically promote or inhibit SUMO modification of just one particular protein.

---

Wang et al., 2014). Our lab and others find that sumoylation represents an important, ligand-independent mode to regulate the NR5A nuclear receptor subfamily that includes Steroidogenic Factor 1 (NR5A1/SF-1) (*Campbell et al., 2008*; *Lee et al., 2011a*; *Lee et al., 2005*) and Liver Receptor Homolog 1 (NR5A2/LRH-1) (*Chalkiadaki and Talianidis, 2005*; *Stein et al., 2014*; *Venteclef et al., 2010*; *Ward et al., 2013*; *Yang et al., 2009*). For instance, knocking-in an unsumoylatable or SUMO-less mutant SF-1 allele leads to profound changes in endocrine physiology and tissue development (*Lee et al., 2011a*). Importantly, even in the presence of one wild-type (WT) allele, the phenotypic effects of SUMO-less SF-1 dominate. Mechanistically, we find that SUMO-less SF-1 can regulate select downstream targets, which we refer to as SUMO-sensitive (*Campbell et al., 2008*). Indeed, sonic hedgehog (SHH) signaling is ectopically activated after expressing SUMO-less SF-1 in both cells and tissues. This gain-of-function or dominance of the SUMO-less SF-1 mutant leads to expansion of select cell types and hormone imbalance, illustrating how disrupting the normal cycle of substrate sumoylation results in disease states. Consistent with our findings, gain-of-function heterozygous missense mutations in the sumoylation site of the transcription factor microphthalmia-associated transcription factor are tightly linked with some forms of human melanoma (*Bertolotto et al., 2011*). SUMO-less variants of the androgen (AR) and glucocorticoid hormone (GR) receptors are also reported to activate new transcriptional programs linked to cellular proliferation (*Paakinaho et al., 2014*; *Sutinen et al., 2014*). These studies suggest that successful efforts to chemically target substrate sumoylation could be used to alter transcription factor activity, to either promote or attenuate SUMO-sensitive genetic programs.

Thus far, efforts to drug sumoylation using in vitro target-based assays and in silico screens have identified different classes of SUMO inhibitors with $IC_{50}$s in the micromolar range (*Table 1*). In vitro target-based screens rely exclusively on defined components (E1, UBC9 and a test substrate), and as such, exclude the contributions by E3s and other unidentified obligate cofactors on substrate sumoylation. An in situ cell-based screen in permeabilized, fixed cells partially overcomes this limitation but still requires the addition of exogenous SUMO components for the assay (*Hirohama et al., 2013*). Currently, inhibitors that target E1 include ginkgolic acid (GA), davidiin, and kerriamycin B. Inhibitors of UBC9 include 2-D08, GSK145A, and spectomycin B1. While GA remains the most widely used and commercially available chemical probe targeting general sumoylation, its efficacy as an inhibitor

**Table 1.** List of sumoylation inhibitors identified by screen type and reported $IC_{50}$ values.

| Compound | Class | Screen | Library | Assay | Substrate | Target | $IC_{50}$ (μM) | Reference |
|---|---|---|---|---|---|---|---|---|
| 2-D08 | Flavonoid | Target | 500 Flavones | IVS | AR Peptide | UBC9 | 6.0 | (*Kim et al., 2013*) |
| Davidiin | Ellagitannin | Target | 750 Extracts | In Situ | RanGap1 | E1 | 0.15 | (*Takemoto et al., 2014*) |
| Ginkgolic acid | Alkylphenol | Target | 500 Extracts | In Situ | RanGap1 | E1 | 3.0 | (*Fukuda et al., 2009a*) |
| GSK145A | Diamino-pyrimidine | Target | GSK Library | IVS | TRPS1 Peptide | UBC9 | 12.5 | (*Brandt et al., 2013*) |
| Kerriamycin B | Antibiotic | Target | 1800 Broths | In Situ | RanGap1 | E1 | 11.7 | (*Fukuda et al., 2009b*) |
| Spectomycin B1 | Antibiotic | Target | Chemical Library | In Situ | RanGap1 | UBC9 | 4.4 | (*Hirohama et al., 2013*) |
| C#21 | Phenyl Urea | Virtual | Maybridge | Docking | RanGap1 | E1 | 14.4 | (*Kumar et al., 2013*) |
| Tannic acid | Gallotannin | Phenotypic | Pharmakon | qPCR | hLRH-1 | E1 | 12.8 | (This Study) |

can vary greatly depending on the assay and substrate being tested (*Bogachek et al., 2014*; *Kim et al., 2013*; *Tossidou et al., 2014*).

Phenotypic cell-based screens offer an alternative approach to in vitro target-based screens for finding new molecular entities (*Swinney and Anthony, 2011*). Here, we set out to identify nontoxic chemical probes that would modulate substrate sumoylation using a cell-based gene-expression screen, which assayed two human LRH-1 (hLRH-1) SUMO-sensitive transcripts as the primary read-out. hLRH-1 is an ideal test substrate for evaluating any potential hits because one has the ability to test how candidate hits affect hLRH-1 activity in both immortalized hepatocellular carcinoma cells and in primary mouse hepatocytes (*Lee et al., 2011b*; *Mataki et al., 2007*; *Oosterveer et al., 2012*). In addition, as shown in this study, hLRH-1 is well-sumoylated in vitro, in cells, and in vivo. From the initial phenotypic screen of the FDA- and European-approved Pharmakon 1600 drug library, the commercial plant extract, tannic acid (TA) was identified as a nontoxic general sumoylation inhibitor, which was effective in multiple platforms, including primary mouse hepatocytes.

## Results

### A phenotypic screen assaying SUMO-sensitive genes identifies TA as the top hit

Sumoylation of hLRH-1 occurs primarily in the flexible hinge domain on two major conserved acceptor lysines K192 and K270, with a minor site located in the N-terminal region at K44 (*Figure 1A*). Similar to our prior results obtained with SF-1, hLRH-1 is efficiently sumoylated (~30%) in human placental choriocarcinoma JEG3 and hepatocellular carcinoma HepG2 cells expressing Flag-hLRH-1 (*Figure 1A*). Importantly, multiple sumoylated hLRH-1 species are readily detected with only the endogenous SUMOylation machinery and without the need to add exogenous SUMO or UBC9. Substituting lysines K192 and K270 with arginines (K192R/K270R or 2KR) eliminates nearly all sumoylated LRH-1 species, as previously noted (*Chalkiadaki and Talianidis, 2005*; *Lee et al., 2005*; *Yang et al., 2009*) and *Figure 1—figure supplement 1*. In vivo sumoylation of hLRH-1 is also equally robust, as observed in mouse liver humanized to express wild-type hLRH-1 (WT) (*Figure 1B*), following viral-mediated infection with recombinant adeno-associated virus serotype 8 (AAV8) (*Cotugno et al., 2012*; *Ill et al., 1997*) and *Figure 1C,D—figure supplement 2*. Impressively, the extent and pattern of hLRH-1 sumoylation in mouse liver are identical to those found in cultured cell lines, demonstrating for the first time that hLRH-1 is efficiently sumoylated at multiple lysines in vivo. By contrast, expressing SUMO-less hLRH-1 (2KR) eliminates nearly all hLRH-1 sumoylation (*Figure 1B*). These collective data establish that hLRH-1 is robustly sumoylated in several platforms, making it an excellent test substrate for assessing both the biochemical and functional effects of small molecule inhibitors of sumoylation.

Rather than assessing substrate sumoylation directly, we used a cell-based screen that monitored two SUMO-sensitive transcripts as the endpoint (diagrammed in *Figure 2A*). The JEG3 cell line was initially chosen because it performed well in all steps of the primary screen and has been used previously to study NR5A activity (*Campbell et al., 2008*). Profiling was carried out on JEG3 cells stably

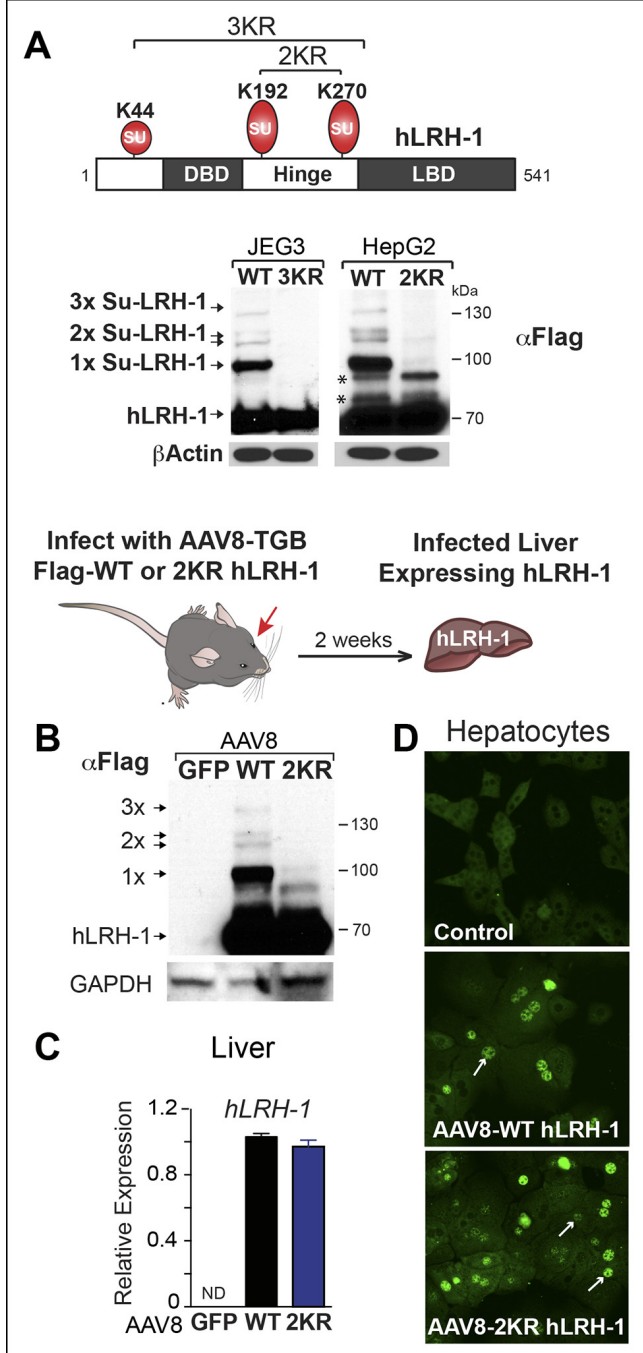

**Figure 1.** Human LRH-1 is efficiently sumoylated in cells and in vivo. (**A**) Schematic of hLRH-1 protein (NR5A2 isoform 1) showing the location of major sumoylation sites at K192 and K270, and the minor K44 site (top panel). WT and SUMO-less forms of hLRH-1 (3KR and 2KR) expressed in JEG3 and HepG2 cells are indicated as detected with anti-Flag antibody. Unsumoylated (hLRH-1) as well as sumoylated hLRH-1 species (1x, 2x, and 3x) are indicated in bottom panel by arrows. Additional bands observed in HepG2 cells that persist after mutating both K192 and K270 are indicated with asterisk (\*). Strategy used to humanize mouse liver for expression of wild type or SUMO-less (2KR) hLRH-1. (**B**) Sumoylated hLRH-1 species detected by anti-Flag in harvested livers after first infecting with AAV8-vectors expressing eGFP, WT or 2KR. (**C**) Relative transcripts levels of hLRH-1 transcripts in mouse livers infected with recombinant AAV8-vectors expressing eGFP, wild-type hLRH-1 (WT) or SUMO-less hLRH-1 (2KR). (**D**) Staining for tagged-hLRH-1 as detected by immunofluorescence using anti-Flag (white arrows). Hepatocytes are prepared as described in 'Materials and methods' from harvested, perfused livers 2 weeks post retro-orbital viral-mediated infection.

*Figure 1 continued on next page*

Figure 1 continued

The following figure supplements are available for figure 1:

**Figure supplement 1.** Mutating individual acceptor lysines in hLRH-1 establishes the importance of K192 and K270 in SUMO modification of hLRH-1.

**Figure supplement 2.** Human LRH-1 transcripts and protein are expressed in liver after AAV8-TBG viral infection.

expressing hLRH-1 or the SUMO-less hLRH-1 mutant, or after siRNA knock down of UBC9 (siUBC9) to identify the most robust SUMO-sensitive genes. To ensure that minor sumoylation on hLRH-1 was eliminated we used the 3KR mutant that disrupts K44, as well as the two major acceptor lysines in the hinge region. Two genes, *APOC3* and *MUC1*,were identified by microarray as the readout transcripts for the screen. These genes are highly induced by either SUMO-less LRH-1 or siUBC9 (*Figure 2B*) and were chosen as two SUMO-sensitive genes in the primary screen assay. Interestingly, *APOC3* can be directly regulated by LRH-1 (*Hwang-Verslues and Sladek, 2008*), whereas *MUC1* is regulated by the androgen receptor (*Rajabi et al., 2011*). In contrast, expression of a well-known NR5A downstream target gene, *CYP11A1* is unaffected by both SUMO-less LRH-1 and siUBC9 knockdown and is thus designated as a SUMO-insensitive LRH-1 target (*Figure 2B*).

A gene-expression-based screen adapted from (*Arany et al., 2008*) assayed *APOC3* and *MUC1* with a 1600-compound drug library (Pharmakon 1600). JEG3 hLRH-1 cells were cultured in a 384-well format, treated with 10 µM of each drug, and measured for *APOC3* and *MUC1* transcripts. Robust Z-scores for each drug treatment were obtained by normalizing the amplification cycle number ($C_T$) of *APOC3* or *MUC1* to *TBP* as an internal control ($\triangle C_T$), and then to the DMSO external control ($\triangle\triangle C_T$). A scatter plot of Z-scores ± 2SD from the primary screen shows 13 drugs producing significant changes in *MUC1* and *APOC3* expression (up and down), representing a 0.8% hit rate (*Figure 2C* and *Source data 1*). Drugs resulting in cytotoxicity or changing *TBP* expression were then filtered out leaving six potential hits listed in *Table 2*.

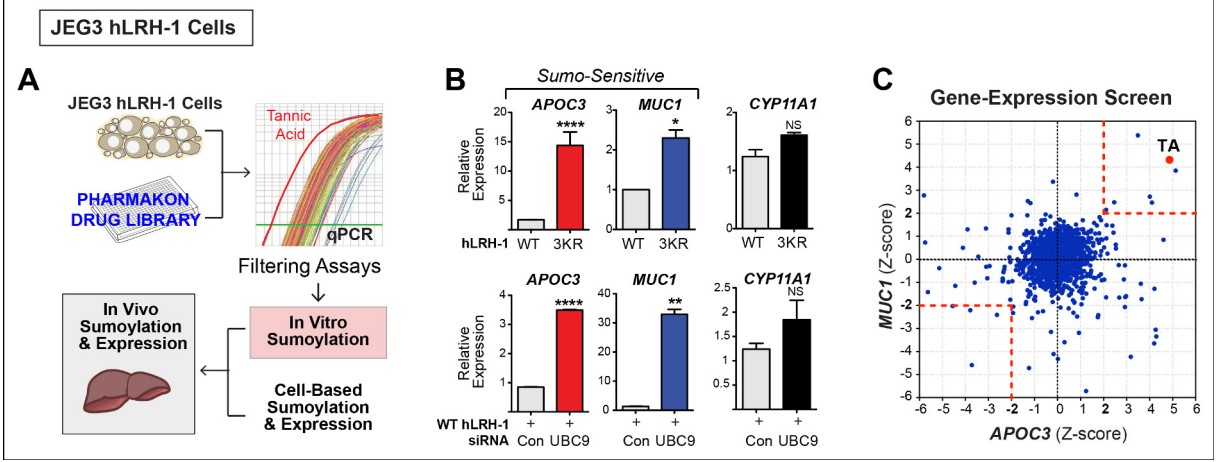

**Figure 2.** A Phenotypic screen identifies TA as a small-molecule sumoylation modulator. (**A**) Schematic outline of the primary screen using JEG3 cells expressing wild-type hLRH-1 and downstream filtering steps to identify small molecules that modulate sumoylation. Individual amplification profiles for *MUC1* transcripts are shown for each drug treatment using the Pharmakon 1600 library (upper right panel). Highlighted in red is the amplification curve of *MUC1* obtained with TA, the top hit from the primary screen. (**B**) Relative levels of SUMO-sensitive transcripts *APOC3, MUC1* and the SUMO-insensitive transcript *CYP11A1* in JEG3 cells expressing wild-type hLRH-1 or SUMO-less hLRH-1 (3KR) (top panel). Relative levels of transcripts as above are shown after 72 hr siControl (Con) or siUBC9 (UBC9) treatment in JEG3 cells expressing hLRH-1. Results represent values obtained for triplicate samples. Statistical significance: ****p<0.0001, **p<0.01, *p<0.05. (**C**) Scatter plot from the primary screen showing normalized Z-scores for *APOC3* and *MUC1*, calculated as described in 'Materials and methods'. All compounds yielding Z-scores greater than +2 or less than -2 are shown within red dashed boxes. Positive Z-scores correspond to increased expression of transcripts relative to the control housekeeping gene, *TBP*. The Z-score obtained for TA is indicated as red dot. TA: Tannic acid.

**Table 2.** List of top six hits from primary screen (Z-scores).

| Drug | APOC3 | MUC1 |
|------|-------|------|
| Tannic acid | 4.87 | 4.31 |
| Trifluridine | 4.01 | 2.71 |
| Taxol | 2.93 | -0.26 |
| Vincristine | -3.61 | -1.01 |
| Colforsin | 1.24 | -5.70 |
| Ouabain | -5.14 | -0.38 |

## TA is a potent inhibitor of substrate sumoylation in cells and in vitro

All potential nontoxic candidates were repurchased and tested in a dose response for induction of the two SUMO-sensitive genes used in the primary assay. Only TA emerged as a valid hit from the primary screen, showing significant induction of *APOC3* and *MUC1*, but not *CYP11A1* (*Figure 3A*). We then asked if hLRH-1 sumoylation is required to observe the stimulatory effects of TA on *APOC3* and *MUC1*. As expected, TA had no effect on *CYP11A1* regardless of the hLRH-1 variant tested (*Figure 3B*). On the other hand, the dose-dependent effects of TA on *APOC3* are lost when tested with the SUMO-less hLRH-1 (3KR), demonstrating that activation of *APOC3* by TA depends on the ability of hLRH-1 to be sumoylatable. Further, *APOC3* upregulation by TA depends on hLRH-1; TA failed to change *APOC3* alone (*Figure 3—figure supplement 1*). On the other hand, the effects of TA on *MUC1* levels are largely independent of hLRH-1, implying that TA can act more broadly on non-hLRH-1 SUMO-sensitive targets.

The transcriptional effects of TA correlated well with the dose-dependent inhibition of hLRH-1 sumoylation in cells, as evidenced by a ~40–50% decrease in all hLRH-1 sumoylated species at 10 µM TA (*Figure 3C*). Diminished hLRH-1 sumoylation is observed only after TA and GA treatment; the other published sumoylation inhibitor, 2′,3′,4′-trihydroxyflavone (2-D08) (*Kim et al., 2013*), and another top screening hit, trifluridine (Tri), failed to show similar results (*Figure 3D*). However, while GA appears more effective than TA at reducing hLRH-1 sumoylation in JEG3 cells (*Figure 3D*), this compound leads to significant cytotoxicity beginning at 10 µM after 5 hr (*Figure 3E*) and 24 hr exposure (*Figure 3—figure supplement 2*). By contrast, TA is nontoxic even at higher concentrations.

Next, we asked whether TA would also inhibit IVS of recombinant full-length human LRH-1 (FL-hLRH-1) protein. In our assay conditions, the pattern of sumoylated FL-hLRH-1 in vitro is identical as that found in cells and in vivo, and collapses down to a single unmodified band after addition of SENP1 (*Figure 4A* and refer back to *Figure 1*). Using IVS conditions that achieve ~50% sumoylation of FL-hLRH-1, TA is the most effective sumoylation inhibitor when compared to 2-D08 and GA, as well as other candidate hits, (*Figure 4B*), with an apparent IC$_{50}$ of 12.8 µM (*Figure 4C* and *Figure 4—figure supplement 1*). As predicted from our initial cellular data, TA inhibits and impairs the rate of sumoylation of other substrates, including recombinant hinge-LBD SF-1, full-length IκBα, and an AR peptide (*Figure 4—figure supplement 2*). As found for davidiin, another tannin sumoylation inhibitor (*Takemoto et al., 2014*), TA impairs E1 thioester formation in non-reducing conditions (*Figure 4D*).

Given that TA and other polyphenols are prone to aggregate formation which can lead to non-specific or promiscuous inhibition (*Feng and Shoichet, 2006*), we assessed the performance of TA in the presence of a non-ionic detergent (Triton X-100), which limits aggregate formation (*Pohjala and Tammela, 2012*). Even in the presence of 0.01% Triton X-100, FL-LRH-1 sumoylation was inhibited by TA treatment, as shown in *Figure 4E*. That TA maintains its inhibitory activity with this detergent suggests that in these assay conditions, the inhibitory effects of TA are not due to non-specific aggregate formation.

Next, we tested whether TA is also effective in hepatocellular carcinoma HepG2 cells, a relevant cell line for studying hLRH-1 (*Figure 5A*). When compared to a nontoxic dose of GA (10 µM) or 2-D08, TA is much more efficient at decreasing levels of sumoylated hLRH-1 and shows little cytotoxicity in HepG2 cells after longer exposure times (*Figure 5B*). We also directly compared the effects of TA versus knockdown of UBC9 (siUBC9) on hLRH-1 and other sumoylated proteins in HepG2 cells.

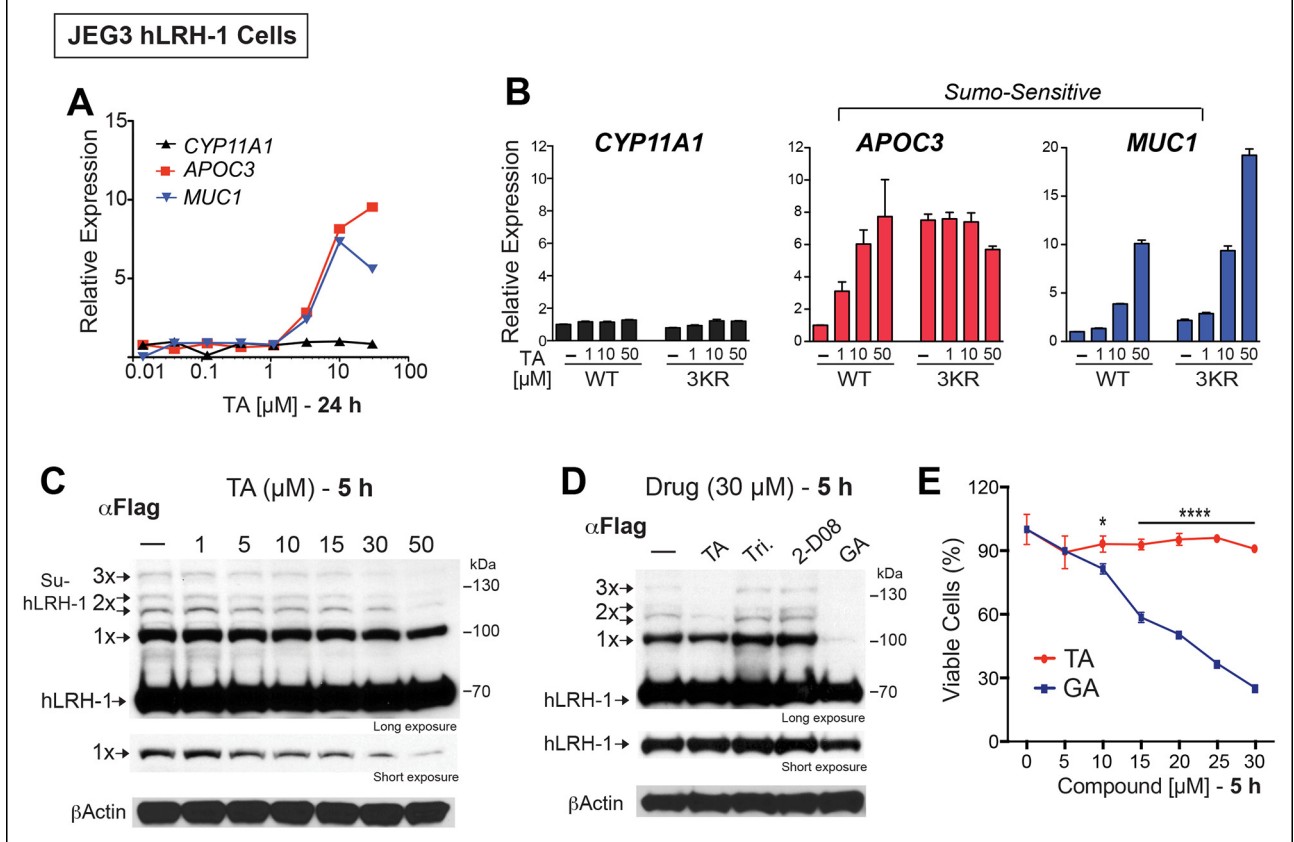

**Figure 3.** TA enhances SUMO-sensitive gene expression in cells. (**A**) Relative expression of *CYP11A1, APOC3*, and *MUC1* in JEG3 cells expressing wild-type (WT) hLRH-1 following 24 hr treatment with increasing TA. (**B**) Relative expression of *CYP11A1, APOC3*, and *MUC1* in JEG3 cells transiently expressing WT or SUMO-less hLRH-1 (3KR) following 6 hr treatment with increasing concentrations of TA as indicated. Vehicle DMSO control is shown (-). (**C**) Levels of unsumoylated (hLRH-1) and sumoylated hLRH-1 species in JEG3 cells detected with anti-Flag following 5 hr treatment with increasing TA concentrations (1–50 μM). Shorter exposure of 1x-sumoylated hLRH-1 is shown in bottom panel (1×). Vehicle control (DMSO) is shown (-). (**D**) Levels of unsumoylated (hLRH-1) and sumoylated hLRH-1 species in JEG3 cells detected with anti-Flag after TA, trifluidine (Tri.), 2-D08, and ginkgolic acid (GA) treatment (30 μM each) following 5 hr treatment. Shorter exposure of unsumoylated hLRH-1 (hLRH-1) is shown in bottom panel. (**E**) Cell viability is shown for JEG3 cells expressing WT hLRH-1 following 5 hr treatment with increasing GA or TA. Statistical significance: ****$p<0.0001$, *$p<0.05$.TA: Tannic acid.

The following figure supplements are available for figure 3:

**Figure supplement 1.** Upregulation of *APOC3* by TA in JEG3 cells depends on presence of hLRH-1.

**Figure supplement 2.** Significant cell toxicity in JEG3 wtLRH-1 cells after 24 hr treatment with GA but not TA.

Surprisingly, despite a substantial loss of UBC9 transcripts (95%) and protein (60%) following siRNA-mediated knockdown for 72 hr (*Figure 5—figure supplement 1*), hLRH-1 remained fully sumoylated (*Figure 5C*). On the other hand, levels of hLRH-1 sumoylation decreased in a dose-dependent manner with TA (*Figure 5C*).The higher migrating sumoylated hLRH-1 species in HepG2 cells that diminish with TA were authenticated as SUMO1 or SUMO2 species (*Figure 5D*). Interestingly, 1x-Su-hLRH-1 appears to be exclusively modified by SUMO1, whereas higher Su-hLRH-1 species are modified by SUMO2. Both siUBC9 and TA inhibit global sumoylation, but in this setting, siUBC9 is slightly more effective (*Figure 5E*). As expected, neither TA nor siUBC9 changes the pool of ubiquitinated proteins. Lastly, we wanted to know if TA similarly modulates sumoylation of endogenously expressed LRH-1. Unfortunately, our ability to cleanly detect or efficiently pulldown endogenous sumoylated LRH-1 species in cells/tissues is difficult with available anti-LRH-1 antibodies, including both commercial and non-commercial sources. Instead, we used human adrenal carcinoma H295R

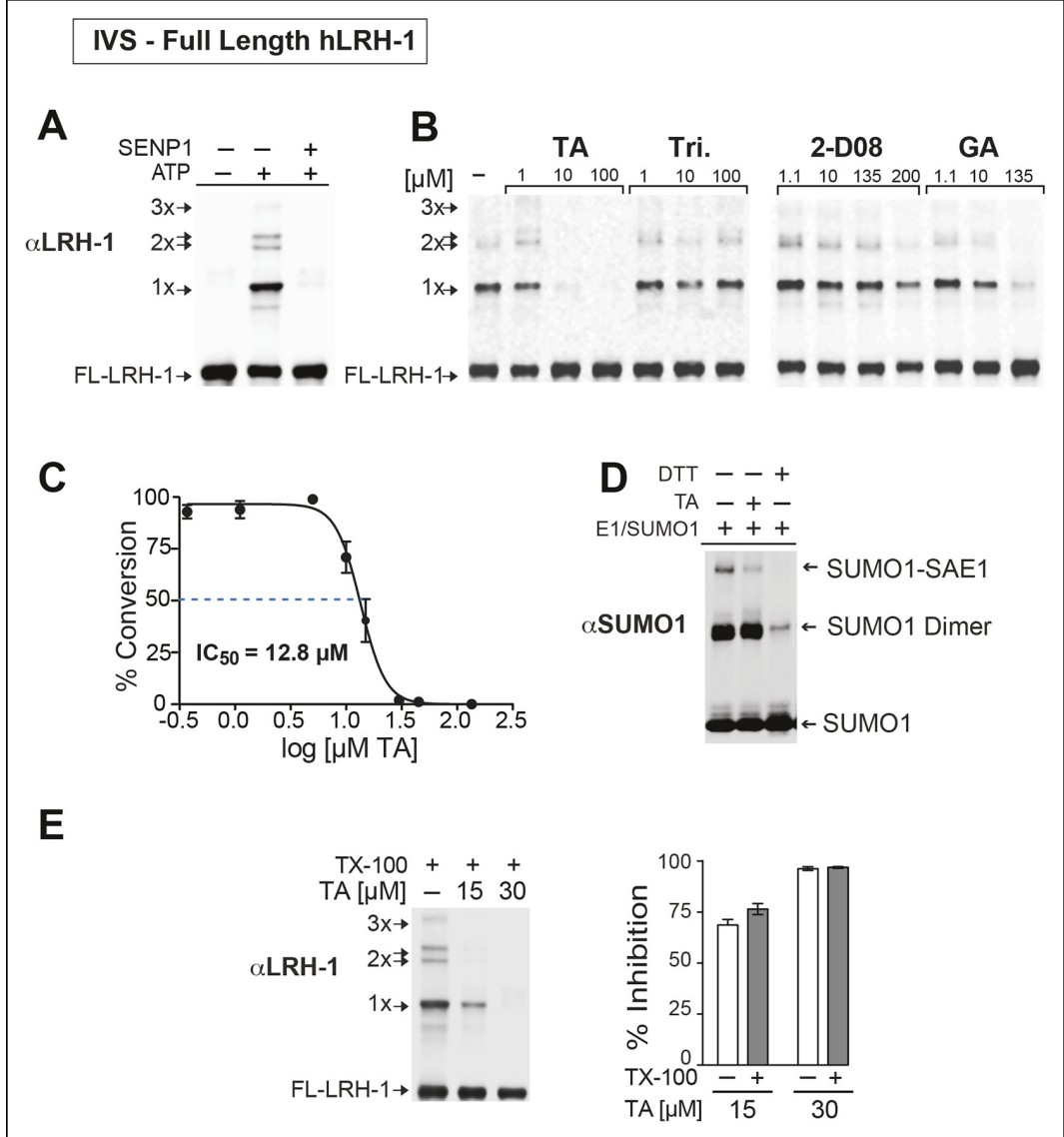

**Figure 4.** TA is a detergent-resistant inhibitor of substrate sumoylation in vitro. (**A**) In vitro sumoylation (IVS) of recombinant full length (FL)-hLRH-1, without ATP, with ATP or with ATP and recombinant SENP1 added to IVS reactions as described in 'Materials and methods'. (**B**) IVS assays with increasing tannic acid (TA), trifluidine (Tri.), 2-D08, and ginkgolic acid (GA). Sumoylated and unsumoylated FL-hLRH-1 are indicated with arrows as detected with anti-LRH-1 antibody. (**C**) $IC_{50}$ of TA in FL-hLRH-1 IVS assay. Data are represented as mean ± SEM from at least three independent replicates. (**D**) Formation of E1 thioester with or without TA, in non-reducing conditions (-DTT). Effects of TA (10 µM) on formation of SUMO-E1 complex (SUMO-SAE1, top band) compared to reducing conditions without TA (-DTT, last lane). SUMO1 dimers are formed in non-reducing conditions (SUMO1 Dimer). E1 thioester formation assays are initiated by addition of freshly prepared ATP (10 mM) and described in 'Materials and methods'. Anti-SUMO1 antibody was used to detect SUMO1 species. (**E**) Levels of sumoylated and unsumoylated FL-LRH-1 in IVS assay with TA (15 and 30 µM) and in the presence or absence of Triton X-100 in vitro (left panel). Bar graph of quantified data showing percent inhibition of hLRH-1 sumoylation by TA and with or without Triton X-100 (right panel).

The following figure supplements are available for figure 4:

**Figure supplement 1.** Effects of TA, other candidate hits, and published sumoylation inhibitors in an IVS assay of full-length hLRH1.

**Figure supplement 2.** IVS of multiple substrates inhibited by TA.

cells that express high levels of endogenous hSF-1 and found that similar to exogenously expressed hLRH-1, TA decreases levels of sumoylated hSF-1 in H295S cells (*Figure 5F*). Similar results were

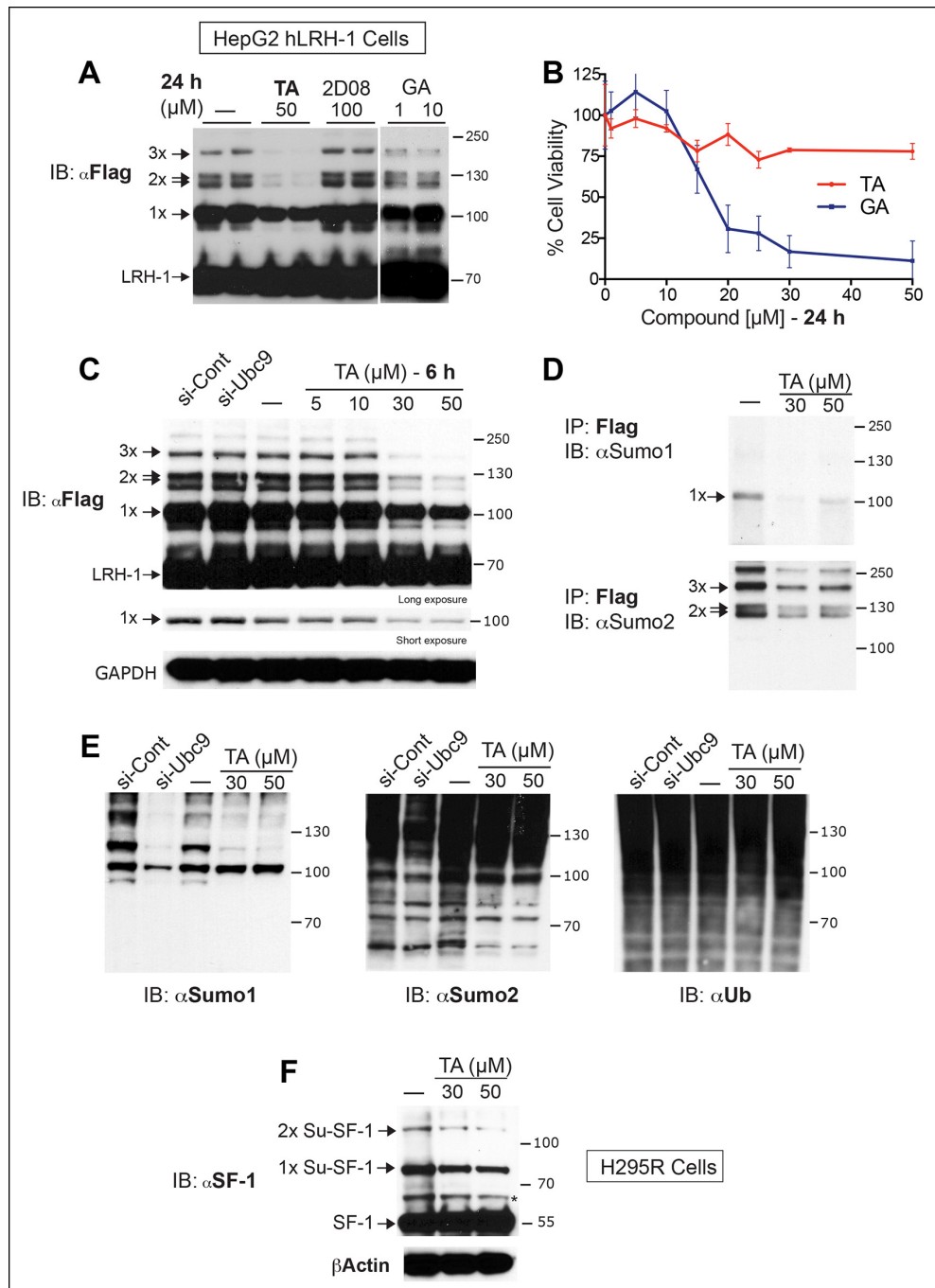

**Figure 5.** TA inhibits exogenous and endogenous NR5A sumoylation, as well as general sumoylation in cells. (**A**) Levels of unsumoylated (LRH-1) and sumoylated hLRH-1 species in HepG2 cells detected with anti-Flag following TA, 2-D08, or GA at specified concentrations after 24 hr treatment. Vehicle control (DMSO) is shown (-). (**B**) Cell viability for HepG2 cells expressing wild-type hLRH-1 following 24 hr treatment with increasing concentrations of GA or TA. (**C**) Sumoylation of wild-type hLRH-1 expressed in HepG2 cells and detected with anti-Flag after siCont or siUBC9 knockdown for 72 hr (+), and with increasing TA (6 h). Shorter exposure of 1x-sumoylated hLRH-1 is shown in panel below (1x), as well as loading control (GAPDH). Vehicle control (DMSO) is shown (-). (**D**) Flag-tagged hLRH-1 protein immunoprecipitated by anti-Flag in HepG2 cells treated with vehicle or TA (30 or 50 µM) followed by immunoblot with either anti-SUMO1 (top panel) or anti-SUMO2 (bottom panel). Arrows indicated migration of 1x, 2x, and 3x sumoylated hLRH-1 species. (**E**) Levels of total sumoylated or ubiquitinated proteins in HepG2 cells following siControl and siUBC9 (72 hr) or TA treatment (6 hr), as detected by anti-SUMO1, -SUMO2, or -ubiquitin. (**F**) Effects of TA (6 hr) on endogenous SF-1 sumoylation in H295R cells and detected anti-SF-1 antibody. GA: Ginkgolic acid; TA: Tannic acid.

The following figure supplements are available for figure 5:

*Figure 5 continued on next page*

*Figure 5 continued*

**Figure supplement 1.** UBC9 transcripts and protein levels following siUBC9 knockdown in HepG2 hLRH-1 cells.
**Figure supplement 2.** TA attenuates endogenous RanGap sumoylation.

also obtained for endogenous RanGap (*Figure 5—figure supplement 2*). Taken together, these data show that TA is a nontoxic potent global sumoylation inhibitor.

## TA enhances hLRH-1 activity and alters SUMO-sensitive targets in HepG2 cells

To ask if TA alters SUMO-sensitive targets in HepG2 cells, we identified a small subset of target genes that is (1) upregulated by hLRH-1, (2) bound by LRH-1 as detected by chromatin-immunoprecipitation high-throughput sequencing (ChIP-Seq) and (3) altered by TA in the presence of hLRH-1 after profiling HepG2 cells (*Source data 2*). Of the 42 genes in this small subset, we identified three genes that were considered SUMO-sensitive and also found to be putative LRH-1 target genes by ChIP-Seq analyses (*Source data 2*), *CYP24A1*, *PFKFB3* and *SERPINE1* (*Figure 6A, B*). TA regulates all three genes in a dose-dependent manner (*Figure 6C*). Moreover, using ChIP-qPCR we find a significant increase in hLRH-1 occupancy on these LRH-1 binding sites, as shown for a site in the *SERPINE1* proximal promoter region and also for an intronic site in *CYP24A1* (*Figure 6D*). These cellular data suggest that TA modulates and increases recruitment to SUMO-sensitive hLRH-1 target genes in a relevant HepG2 cellular model system.

## TA functionally mimics SUMO-less LRH-1 in vivo

We then determined if TA functions as a nontoxic sumoylation inhibitor in primary mouse hepatocytes and if TA is able to recapitulate gene expression changes observed with SUMO-less hLRH-1. Limited profiling was carried out to identify SUMO-sensitive genes in infected mouse liver overexpressing WT or SUMO-less hLRH-1 (2KR). Both WT and SUMO-less hLRH-1 were expressed equivalently with levels ~10-fold higher than that of endogenous mLRH-1, as judged by immunoblots using an anti-LRH-1 antibody that detects both mouse and hLRH-1 (*Figure 7A*). Expression of endogenous *mLrh*-1 is unaffected after infecting mice with AAV8-hLRH-1 (refer back to *Figure 1—figure supplement 2*). Although WT and SUMO-less hLRH-1 upregulate the classic LRH-1 target *Cyp8b1* (*Lee et al., 2008*), expression of SUMO-less hLRH-1 leads to robust activation of *adiponectin (Adipoq)* and *sonic hedgehog (Shh)*, in mouse liver (*Figure 7B*). The pattern of hLRH-1 sumoylation is preserved in cultured hepatocytes, but lost in primary cultures expressing the SUMO-less hLRH-1 mutant (*Figure 7C*, left panel). Importantly, when tested in primary mouse hepatocytes TA (10 µM, 5 hr) diminishes levels of sumoylated WT hLRH-1, with nearly all sumoylated species absent including 1x-SUMO-hLRH-1 (*Figure 7C*, right panel); thus closely resembling the SUMO-less 2KR mutant.

SUMO-sensitive genes identified in whole liver were then tested with TA in mouse primary hepatocytes that express only endogenous mLRH-1. As found in the liver, *Shh*, its downstream effector *Gli2*, and *Adipoq* are essentially switched on by TA (*Figure 7D*). This trigger-like effect of TA and SUMO-less hLRH-1 on *Shh* and *Gli2* is consistent with our previous findings that hedgehog signaling can be ectopically activated in mouse organs after knocking-in SUMO-less SF-1 (*Lee et al., 2011a*). Importantly, cell viability was unchanged by TA after 24 hr exposure (*Figure 7—figure supplement 1*). These data establish that in primary cells, TA abrogates hLRH-1 sumoylation and induces hLRH-1 SUMO-sensitive downstream target genes.

## Discussion

Here, using a phenotypic, gene-expression-based screen we identify the polyphenol, TA as a potent inhibitor of hLRH-1 sumoylation in multiple platforms. In vitro assays confirm that TA impairs substrate sumoylation. When tested n several cellular model systems, an acute, nontoxic dose of TA treatment markedly reduces levels of sumoylated hLRH-1, as well as hSF-1, and modulates expression of hLRH-1 target genes. Impressively, in primary cultures of mouse hepatocytes, TA inhibits

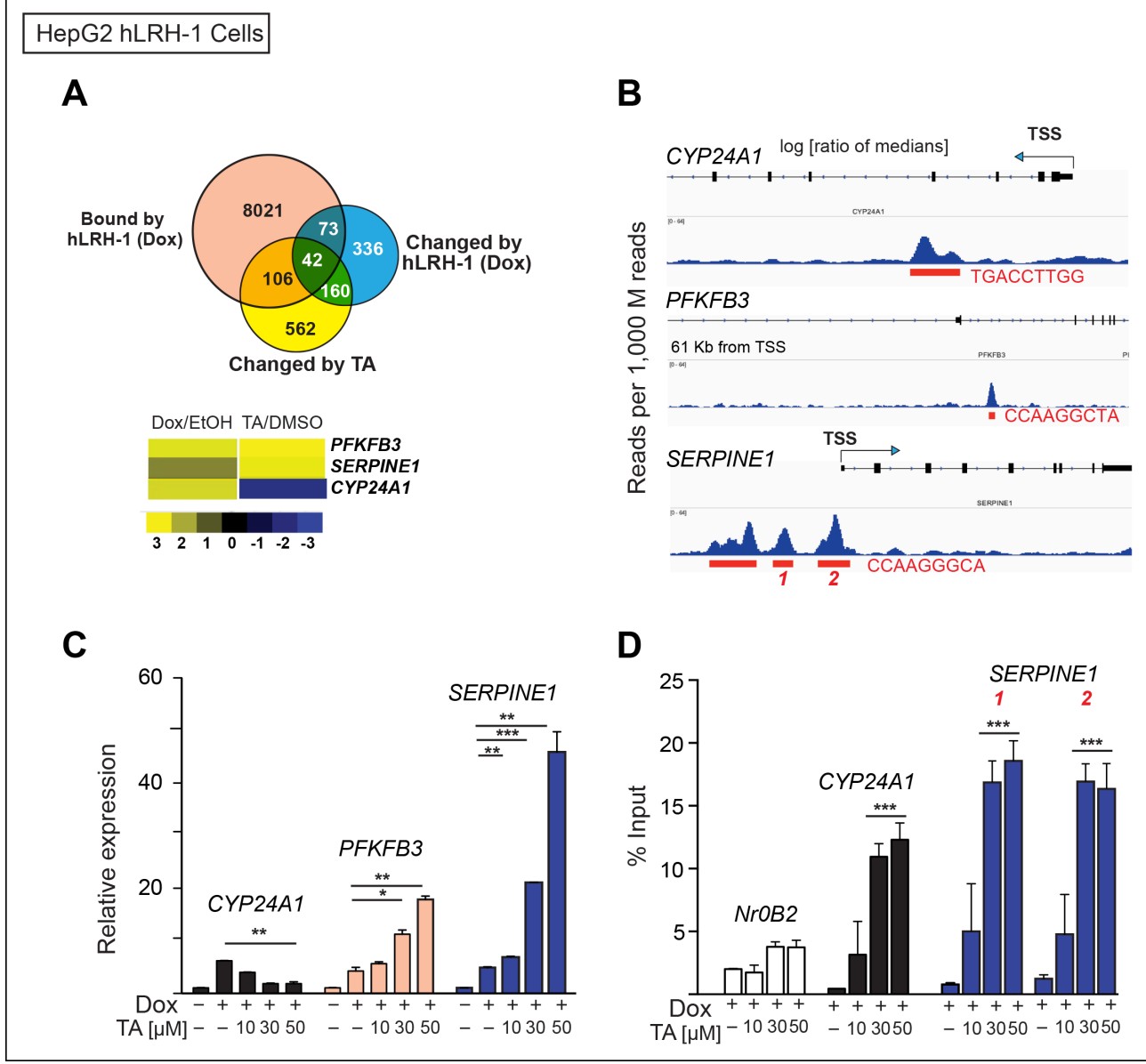

**Figure 6.** TA increases expression and promotes hLRH-1 occupancy on target genes. (**A**) Venn diagram representing the overlap between transcripts changed by induction of hLRH-1 (+Dox) (Blue) and hLRH-1+TA (+Dox, +TA 30 μM) (Yellow), as well as hLRH-1 binding sites identified by ChIP-Seq in HepG2 cells (+Dox, Orange). Heat map of top three genes from overlapping set of 42 genes: *PFKFB3, SERPINE1 (PAI1)*, and *CYP24A1* showing changes after induction of hLRH-1 (Dox/EtOH) and then after TA treatment (TA/DMSO). (**B**) ChIP-Seq binding profiles of the three hLRH-1 targets from panel A. Representative views for ChIP-Seq peaks called by MACS are shown along with genomic location and consensus sequence of putative hLRH-1 binding sites (red text). (**C**) Relative expression of three hLRH-1 targets in HepG2 cells from (**A** and **B**) before (-Dox) and after induction of hLRH-1 (+Dox) and following treatment with TA for 6 hr. (**D**) ChIP-qPCR results in HepG2 cells expressing hLRH-1 for regions identified in panel B, with vehicle control (-) and TA (6 hr). Statistical significance for panels C and D: ***p<0.001, **p<0.01, *p<0.05.

The following figure supplement is available for figure 6:

**Figure supplement 1.** Quantification of transcriptional changes in HepG2 cells after TA or siUBC9 treatment and in presence of hLRH-1.

hLRH-1 sumoylation and is able to mimic the transcriptional output of the SUMO-less LRH-1 mutant by switching on genes that have been associated with hepatic injury.

Our gene-expression-based screen was designed to sample the transcriptional output of substrate sumoylation, which has not been possible with in vitro target-based screens. We reasoned

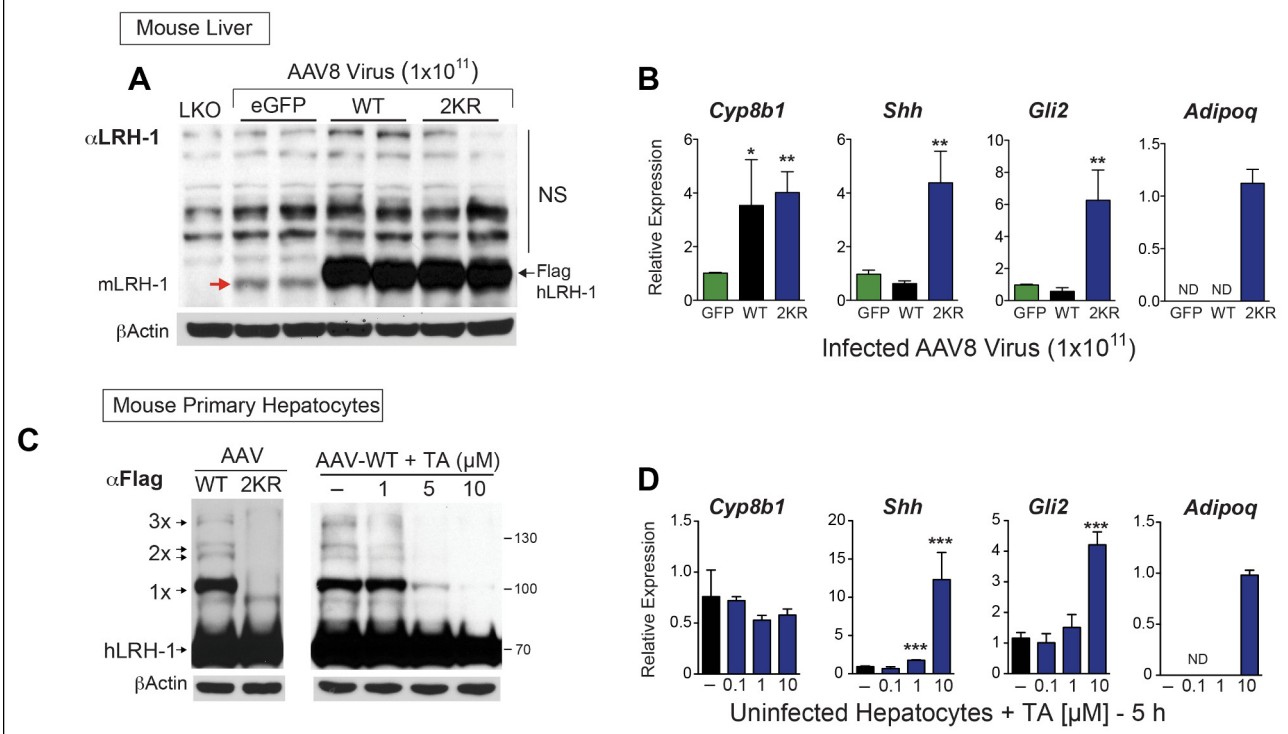

**Figure 7.** TA mimics SUMO-less hLRH-1 when expressed in humanized mouse primary hepatocytes. (**A**) Relative expression of hLRH-1 in mouse livers 2 weeks post-infection with amount of vector indicated (GC/ml). Hepatic expression of endogenous mouse mLRH-1 (red arrow) in control C57BL/6 mice, in *LKO* mice (*mLRH-1^f/f^;Alb-Cre*), or in C57BL/6 mice 2 weeks post infection with recombinant AAV8-eGFP (eGFP), AAV8-WT-hLRH-1 (WT), or AAV8-2KR-hLRH-1 (2KR) (black arrow). Endogenous mLRH-1 and exogenous hLRH-1 are detected with an anti-LRH-1 antibody. Note that non-specific bands detected with the anti-LRH-1 antibody obscure sumoylated LRH-1 species. (**B**) Relative expression in mouse liver of the classic LRH-1 target, *Cyp8b1* or SUMO-sensitive LRH-1 targets *Adipoq* and *Shh* (and its downstream target *Gli2*) following infection with either AAV8-eGFP, AAV8 hLRH-1 or a SUMO-less AAV8-2KR-hLRH-1. Each bar represents values obtained from three livers. Values below the threshold of detection (qPCR >40 cycles) are indicated as ND. (**C**) Sumoylation pattern of AAV8 hLRH-1 (WT) and AAV8-2KR hLRH-1 (2KR) in cultured mouse primary hepatocytes (left panel). Sumoylation of hLRH-1 in infected primary mouse hepatocytes treated with increasing TA for 5 hr (right panel). Sumoylated species and unsumoylated Flag-hLRH-1 are indicated by arrows and detected by anti-Flag antibody. (**D**) Relative expression of genes shown in Panel B measured in uninfected primary mouse hepatocytes treated for 5 hr with increasing concentrations of TA. Statistical significance for panels B and D: \*\*p<0.01, \*p<0.01. TA: Tannic acid.

The following figure supplement is available for figure 7:

**Figure supplement 1.** No cellular toxicity in primary hepatocytes by TA.

that assessing transcriptional endpoints makes it possible to integrate other mechanisms that promote substrate-specific sumoylation or desumoylation, including possible synergistic effects of multiple transcription factor sumoylation (*Holmstrom et al., 2008*; *Komatsu et al., 2004*). Several factors make hLRH-1 an excellent test substrate in initial and follow-up assays. First, our earlier work established that blocking the NR5A sumoylation cycle results in robust transcriptional changes that could be leveraged in a phenotypic screen. Second, hLRH-1 is well-sumoylated with a stereotypic pattern of multiple sumoylated species requiring only the endogenous sumoylation machinery when it is expressed in HEK293S, JEG3, and HepG2 cells, in primary hepatocytes and in liver tissue. These findings differ from the need for exogenous SUMO1/2 and E3s required to detect 1x-sumoylated V5-tagged mLRH-1 in HEK293 cells or primary hepatocytes (*Stein et al., 2014*). In addition, when compared to hFXR, another sumoylated hepatic NR, the number and intensity of hLRH-1 sumoylated species far exceed the faint, single FXR sumoylated band detected when hFXR is expressed in mouse liver (*Kim et al., 2015*). Finally, the pattern of hLRH-1 sumoylation is readily duplicated in vitro, allowing one to also test candidate sumoylation modulators in a defined cocktail. We posit that NR5As might be particularly good substrates for sumoylation because their major acceptor lysines reside in the flexible hinge domain, possibly promoting strong protein–protein interactions

previously observed between Ubc9 and both SF-1 and LRH-1 (*Lee et al., 2005*). Indeed, the residual UBC9 protein following siUBC9 knockdown likely accounts for the persistent levels of 1x SU-LRH-1 and suggests that once UBC9 is charged with SUMO1/2, sumoylation of hLRH-1 proceeds efficiently. Interestingly, as with siUBC9, inhibiting E2 by 2-D08 is also ineffective at blocking hLRH-1 sumoylation. Regardless of the underlying mechanisms that confer relatively high basal levels of NR5A sumoylation in cells and in tissues, the consistent and robust sumoylation observed for hLRH-1 were instrumental in facilitating follow-up studies on candidate small molecule hits from our primary screen.

While polyphenols such as TA are easily dismissed as promiscuous inhibitors and false positives in high-throughput screens (*Feng and Shoichet, 2006*; *Pohjala and Tammela, 2012*), the effects of TA in our sumoylation assays are extremely reproducible in multiple cell lines and in primary hepatocytes. Interestingly, although other colloid-forming pehnolic compounds, such as bergapten and coumarin 153 (*Pohjala and Tammela, 2012*), are present in the Pharmakon library, both failed to emerge as hits in the primary screen. When compared to other sumoylation inhibitors TA performs well. Indeed, in our IVS assay conditions, TA is more effective than either 2-D08 or GA and can inhibit sumoylation of multiple substrates in vitro, including the hinge-LBD of SF-1, an androgen receptor peptide, and full-length IκBα. The fact that 2-D08 fails to inhibit hLRH-1 sumoylation but is effective on other substrates (AR peptide and FL-IκBα) might reflect the fact that 2-D08 fails to block the strong interactions between Ubc9 and NR5As, as mentioned above. These data imply that mechanistically distinct sumoylation inhibitors act on different classes of substrates. We also find that unlike GA, which decreases cell viability as shown here and reported by (*Liu and Zeng, 2009*), TA appears to be well-tolerated in both immortalized and primary cell cultures. Hence, while GA might decrease sumoylation as an adaptive response to cell death, the utility of GA in assessing the transcriptional responses of substrate sumoylation is potentially quite limiting.

Our data suggest strongly that TA blocks substrate sumoylation by inhibiting E1 thioesterization, as found for the ellagitannin, Davidiin (*Takemoto et al., 2014*). The known aggregate formation and antioxidant properties of TA appear to be less important in inhibiting substrate sumoylation. Indeed, TA inhibits FL-hLRH-1 sumoylation even in the presence of detergent. Polyphenols, including TA, are also antioxidants and can scavenge reactive oxygen species (ROS) during oxidative stress (*Chen et al., 2007*; *Yazawa et al., 2006*), which might also directly affect the equilibrium between sumoylation-desumoylation (*Bossis and Melchior, 2006*). In this regard, we find that two other antioxidants, ellagic acid and EGCG, are ineffective at inhibiting hLRH-1 sumoylation (data not shown). Furthermore, conditions in our IVS assays are highly reducing making it unlikely that TA inhibits LRH-1 sumoylation via its antioxidant properties in this setting.

That TA is effective at blocking hLRH-1 sumoylation in humanized primary hepatocytes greatly strengthens the validity of TA as a useful chemical tool to assess the cellular effects of sumoylation. Interestingly, TA is more effective at blocking hLRH-1 sumoylation in primary hepatocytes as compared to HepG2 cells where 1x SUMO-hLRH-1 persists even at the highest dose of TA; a similar trend was noted for endogenous hSF-1 in H295R cells. The lower efficacy of TA in immortalized cell lines may reflect an increase in the general sumoylation machinery in immortalized versus primary cells, as noted by (*Bellail et al., 2014*). The use of humanized mouse hepatocytes and the dramatic changes we observed in *adiponectin* and *sonic hedgehog* transcripts may begin to provide new insights into the in vivo function of LRH-1 sumoylation. The ectopic activation of SHH signaling observed here in primary hepatocytes after overexpressing SUMO-less hLRH-1 and after TA treatment confirms our earlier work showing that elimination of SF-1 sumoylation activates hedgehog signaling in endocrine tissues (*Lee et al., 2011a*). Others have noted that hyperactivation of hedgehog signaling in liver is associated with non-alcoholic steatohepatitis (NASH) progression and responses to liver injury (*Grzelak et al., 2014*; *Guy et al., 2012*; *Hirsova and Gores, 2015*). Adiponectin is an adipocyte-derived protein that reduces fatty liver (*Xu et al., 2003*) and appears protective against NASH (*Asano et al., 2009*). Indeed, while *adiponectin* is normally never expressed in liver, hepatic *adiponectin* transcripts are observed in rats after chemically induced hepatotoxicity (*Yoda-Murakami et al., 2001*) and in patients with fatty liver or fully progressed NASH (*Uribe et al., 2008*). The finding that SUMO-less hLRH-1 and TA switch on hepatic *adiponectin* and hedgehog signaling leads us to speculate that tipping the balance of the hLRH-1 sumoylation cycle toward desumoylation might initiate adaptive responses to liver injury, and eventually a pro-inflammatory response, as suggested by others (*Venteclef et al., 2010*). Interestingly, a global knock-in of a single

SUMO mutation (K289R) in mouse LRH-1, which is equivalent to K270R in hLRH-1, has no strong phenotype on its own, but mitigates aortic plaque formation in $Ldlr^{-/-}$ arteriosclerosis-prone mice (*Stein et al., 2014*). Hence, revealing the full physiological consequences of LRH-1 sumoylation could require the elimination of both major sumoylation sites in the flexible hinge domain and the use of conditional knock-in strategies that are specific for the adult liver.

In summary, using a novel cell-based assay, we report that the commercially derived, plant extract TA is a useful, nontoxic chemical tool for assessing the transcriptional and cellular effects of sumoylation in both immortalized and primary cell cultures. Based on our collective studies that have focused on the sumoylation of NR5As, we propose that the ratio of sumoylated to desumoylated substrate can be chemically manipulated to switch on and off sumo-sensitive transcriptional programs. Clearly, continued efforts are needed to determine whether more selective chemical tools can be found that promote or block sumoylation of a given substrate.

## Materials and methods

### Cell lines and transfections

To generate tetracycline (TET)-inducible Flp-In T-REx stable JEG3 cells, 3x Flag-tagged WT and 3KR (K44R/K192R/K270R) hLRH-1 were cloned into pcDNA5/FRT/TO expression vectors (Life technologies, South San Francisco, CA), followed by selection with 100 or 125 µg/ml Hygromycin B (Gemini Bio-Products, Sacramento, CA). JEG3 hLRH-1 cells were treated with tetracycline (100 ng/ml, Teknova Laboratory, Hollister, CA) for 6 hr to induce WT or SUMO-less LRH-1 proteins.

Doxycycline (Dox)-inducible HepG2 3G stable cells were made by cloning 3x Flag-tagged WT and 2KR (K192/270R) hLRH-1 into pTRE 3G vectors (Clontech, Mountain View, CA), followed by selection with 250 µg/ml Hygromycin B (Gemini Bio Products, Sacramento, CA). The TET-On 3G HepG2 parental cell line was a generous gift from Dr. Stephen Hand (*Li et al., 2012*). For detecting WT or SUMO-less LRH-1 expression, HepG2 3G cells were treated with 200 ng/ml Dox (Sigma-Aldrich, St. Louis, MO) for 6 hr. For siUBC9 knockdowns, Ubc9 (SI04185937, SI04368420) and non-silencing control (SI03650318) siRNA were purchased from Qiagen, Hilden, Germany. SiRNA at 5 nM final concentration was reverse-transfected into JEG3 or HepG2 WT hLRH-1 stable cells by RNAiMax (Life Technologies) for 72 hr followed by induction of hLRH-1 expression by addition of 100 ng/ml TET for 24 hr to JEG3 cells or by addition of 250 ng/ml Dox for 6 hr to HepG2 cells.

### Cell viability assay

For cell viability assays, JEG3 hLRH-1 or HEPG2 hLRH-1 cells were plated on 24-well plates in 0.5 ml of media. Primary hepatocytes were seeded on 96-well plates in 0.1 ml of media. The following day, fresh media was applied with compounds or DMSO control. After 5 or 24 hr treatment, cell viability was assayed using CelltiterGlo (Promega, Madison, WI) according to manufacturer's instructions. Relative luminescence was measured with Veritas Microplate Luminometer (Turner BioSystems, Sunnyvale, CA) with an integration time of 1.0 s and normalized to the DMSO control.

### Immunoprecipitation and western blotting

Cells were lysed in RIPA buffer (6 mM $Na_2HPO_4$, 4 mM $NaH_2PO_4$, 2 mM EDTA pH 8.0, 150 mM NaCl, 1% NaDOC, 0.1% SDS and 1% NP-40) and tissues were lysed in Tris–SDS buffer (2% SDS, 0.6 M Tris-Cl pH 8.0 and 0.1 M DTT) supplemented with protease inhibitors (Sigma-Aldrich) and 20 mM N-Ethylmaleimide (NEM; Sigma-Aldrich) and sonicated using the Diagenode Bioruptor. Lysates were clarified by centrifugation and protein concentration was measured using Protein Assay Dye reagent concentrate (Bio-Rad, Hercules, CA) according to the manufacturer's protocol. The following antibodies and concentrations were used: anti-Flag M2; 1:7500 (Sigma-Aldrich), anti-LRH-1; 1:3000 for mouse liver and 1:10,000 for in vitro assay (R&D, Minneapolis, MN), anti-SF-1; 1:1000 (Upstate, EMD Millipore, Billerica, MA), anti-UBC9; 1:1000 (Cell Signaling, Danvers, MA), anti-SUMO1; 1:1000 (Developmental Studies Hybridoma Bank, Iowa City, IA), anti-SUMO2; 1:1000 (Life technologies) and Ubiquitin monoclonal P4D1 antibody; 1:1000 (Cell Signaling), HRP-conjugated anti-βactin 1:2500 (Cell Signaling), and anti-GAPDH 1:5000 (Santa Cruz Biotechnology, Santa Cruz, CA).

## Quantitative real-time polymerase chain reaction

Total RNA from cells and tissues were isolated using Trizol Reagent (Life Technologies) and PureLink RNA mini kit (Life Technologies), respectively. DNase-treated 1 µg total RNA was used to generate cDNA using High-Capacity cDNA Reverse Transcription kits (Life Technologies). RT-qPCR was performed using SYBR Green, High ROX (Biotool, Houston, TX, or Quanta, Gaithersburg, MD) and data analyzed essentially as described (*Kurrasch et al., 2007*). Sequences for all primer pairs used for qPCR reactions are listed in *Supplementary file 1*.

## Primary screening workflow and gene-expression-based qPCR assays

JEG3 WT hLRH-1 cells (5000 cells per well) were plated into 384-well by cell dispenser Wellmate (Thermo Scientific, Waltham, MA) for 24 hr. Using an FDA- and European-approved Pharmakon library of 1600 compounds (MicroSource Discovery Systems, Gaylordsville, CT), drugs were pinned at a concentration of 10 µM in 0.1% DMSO using a Biomek FXP (Beckman, Pasadena, CA). At the same time, cells were treated with Tet (100 ng/ml) for inducing WT hLRH-1. Twenty-four hours later, cells were washed once in PBS and then lysed in 25 µl of lysis buffer provided in the TurboCapture 384 mRNA Kit (Qiagen) using EL406 microplate washer (BioTek, Costa Mesa, CA). After a 10 min incubation at 37°C, 20 µl of cell lysate was transferred to 384-well oligo (dT)-coated plate (Qiagen) using Biomek FXP and incubated at room temperature for 90 min with shaking. Plates were washed three times with washing buffer and reverse transcription was performed in the same well using High-Capacity cDNA Reverse Transfection kits (Life Technologies), according to the manufacturer's instruction, with a total volume of 20 µl. Aliquots of cDNA was delivered to 384-well qPCR plates using a Biomek FXP Liquid handler and stored at -20°C for subsequent qPCR assays. See *Supplementary file 2* for further details. Retesting of top candidates was carried out with repurchased drugs from sources listed in *Supplementary file 3.*

For qPCR assays, the master-mix buffer (8 µl) containing PCR oligos and Quanta qScript cDNA SYBR Mix (Quanta Biosciences) was added to cDNA (2 µl). All qPCR assays were performed using an ABI 7900HT instrument. Data were analyzed using the $\Delta\Delta C_T$ method. Average of $\Delta CT$ from DMSO-treated samples (N = 58) was used as external controls. *MUC1* and *APOC3* genes were used as the endpoint read-outs for the SUMO-sensitive genes.

Calculations:

$\Delta C_T$ = Gene of interest (*APOC3* or *MUC1*) - $C_T$ House Keeping gene (*TBP*) $C_T$

$\Delta\Delta C_T$ = $\Delta C_T$ (Drug) - $\Delta C_T$ (DMSO)

For Reference: $1 C_T$ change = 2-fold change

**Z-score:** -(Drug $\Delta\Delta C_T$ – Average $\Delta\Delta C_T$ (for all 1600 Drugs)/SD $\Delta\Delta C_T$ (for all 1600 Drugs)

For Reference: Positive Z-score = Upregulation of *MUC1/APOC3*, Negative Z-score = Downregulation of *MUC1/APOC3*

## In vitro sumoylation and thioester assays

Full-length LRH-1 (aa1-541; UniprotKB entry: O00482) was subcloned into pRSF-2 vector (Novagen, Madison, WI) and grown in *Escherichia coli* BL21Star (DE3) cells (Invitrogen) at 16°C for 16–18 hr to an OD 0.6–0.8 and induced with IPTG (0.2 mM). Cells were resuspended in lysis buffer A (20 mM Tris–HCl pH 8, 300 mM NaCl, 10% glycerol, 40 mM Imidazole, 5 mM β-mercaptoethanol [BME], 1 mM CHAPS) supplemented with protease inhibitors (Roche, Indianapolis, IN). hLRH-1 protein was purified using Ni-nitrilotriacetate beads (Qiagen) and eluted with Buffer A and 300 mM imidazole. Eluted hLRH-1 was bound to 24 bp duplex region of the *Inhibin-A* promoter (5'-GGAGA-TAAGGCTCATGGCCACAGA-3') to stabilize protein and was further purified by size exclusion chromatography in Superdex 200 (16/60). Native gel electrophoresis confirmed that the hLRH-1/DNA complex eluted as a monomer.

His-tagged components of sumoylation reactions including hE1, hUBC9, and hSUMO1 were grown to an OD 0.3–0.7 and induced with IPTG (0.35 mM) for 5 hr at 22°C. Cells were lysed in 50 mM Tris–HCl, pH 8.0, 150 mM NaCl, and 5% glycerol supplemented with protease inhibitors and proteins purified as described (*Reverter and Lima, 2009*; *Yunus and Lima, 2009*). FL-hLRH-1 was prepared and lysed in 20 mM Tris–HCl pH 8.0, 1 mM CHAPS, 10% glycerol, 5 mM BME, 20 mM imidazole, and 300 mM NaCl supplemented with protease inhibitors and then eluted with lysis buffer with 300 mM Imidazole. IVS reactions were performed at 37°C for 1 hr using 0.1 µM E1, 10 µM

UBC9, 30 µM SUMO1, and 1 µM FL-hLRH-1 substrate (*Ward et al., 2013*) in 50 mM Tris–HCl, 100 mM NaCl, 10 mM MgCl$_2$, 2 mM DTT and initiated by addition of freshly made 10 mM ATP. Aggregation assays used 0.01% Triton X-100 (Sigma). IVS reactions were quenched with 4x Laemmli Buffer with BME, boiled for 5 min and loaded onto a Novex Nupage 4–12% Bis-Tris gel and transferred to nitrocellulose membranes followed by incubation with mouse anti-LRH-1 (1:7500, R&D) or mouse anti-SUMO1 (1:325, DSHB). Proteins were visualized using LiCor Odyssey system and goat anti-mouse 800 (1:20,000 LiCor, Pierce, Lincoln, NE) and quantitated by Image Studio Lite. Percent conversion was calculated by the ratio of sumoylated protein over total signal per reaction normalized to DMSO control. Concentration curves were derived from at least three independent reactions and fit with nonlinear fitting of log$_{10}$ [µM TA] versus variable slope using Prism graphing software (GraphPad, La Jolla, CA). IVS of full-length IκBα and fluorescent AR peptide was performed as previously described (*Kim et al., 2013*). Conditions for the thioester assay were as described above, but with only E1 and SUMO1 proteins added to IVS reactions.

## Microarrays

Human Exonic Evidence Based Open-source (HEEBO) arrays were printed at the UCSF Center for Advanced Technology (CAT). Hybridization conditions were carried out in Flp-In T-REx JEG3 cells as previously described (*Kurrasch et al., 2007*) to identify top genes upregulated by expression of SUMOless hLRH-1 versus WT hLRH-1, or after Ubc9 knockdown as described above. For siRNA experiments, HepG2 hLRH-1 cells were reverse-transfected with 5 nM of pooled siRNA directed against human siUBC9 or siRNA control from Qiagen, with RNAiMAX transfection reagent (Life technologies) according to the manufacturer's protocol. Seventy-two hours after siRNA transfection, WT hLRH-1 was induced with Dox for 6 hr. Total RNA was purified using RNAeasy kit (Life technologies) according to the manufacturer's protocol. Hybridizations were performed at 65°C for 16 hr using mixers compatible with the MAUI hybridization systems (BioMicro Systems, Salt Lake City, UT). Arrays were scanned using an Axon Scanner 4000B, and data analyzed by GenePix 6.0 software (Molecular Devices). Heat maps were generated using open-access TreeView software.

## Chromatin immunoprecipitation – DNA sequencing

HepG2 hLRH-1 cells were seeded (4 x 10$^6$) on 10 cm plates overnight, induced with 250 ng/ml Dox and treated with DMSO or 50 µM TA for 6 hr. Cells were cross-linked with 1% formaldehyde for 3 min at room temperature and quenched by addition of 400 mM glycine. Cells were harvested in 50 mM HEPES-KOH pH 7.4, 1 mM EDTA, 150 mM NaCl, 10% glycerol, and 0.5% Triton X-100, swelled for 40 min at 4°C, then nuclei were pelleted at 600 x g for 5 min and resuspended in RIPA buffer (10 mM Tris–HCl pH 8.0, 1 mM EDTA, 150 mM NaCl, 5% glycerol, 0.1% sodium deoxycholate, 0.1% SDS, 1% Triton X-100) (*Watson et al., 2013*). Lysates were sonicated for a total of 30 min (30 s on, 30 s off, 5 min intervals) with a Diagenode Biorupter UD-200 on High setting at 4°C. Sonicated chromatin was clarified by centrifugation then IP'd with 1 µg anti-Flag M2 antibody pre-conjugated to 10 µl Protein G Dynabeads (Invitrogen) for 2 hr at 4°C. Bound protein were washed with 500 mM NaCl and LiCl buffer before reverse cross-linking and proteinase K digested overnight. DNA was isolated using Zymogen ChIP DNA Clean and Concentrator columns and pooled for deep sequencing. ChIP DNA was sent to Hudson Alpha Genomic Services Laboratory for library preparation using Illumina TruSeq Kit.

## Bioinformatics analysis

Triplicates of hLRH1 ChIP-Seq (WT hLRH-1) and a control (Input) were sequenced on the Illumina HiSeq 2000 platform using 50 bp, single-end reads. Reads were mapped to the hg19 human reference using bowtie and de-duplicated using Samtools. Final data compilation includes a total of 3.36, 10.77, and 7.7 million aligned sequence reads for WT hLRH-1 and 2.67 million reads from Input. Quality control and ChIP-signal strength assessment was performed via CHANCE (*Diaz et al., 2012*). CHANCE called both experiments as successful (via a comparison with the distribution of ChIP-strengths observed in the ENCODE repository) at a combined, positive false discovery rate (FDR) of FDR = 2.1X10$^{-4}$ for WT hLRH1. Reads from replicates were then pooled, and peaks were called via MACS (*Zhang et al., 2008*), using the default parameter settings. This generated 18,884 peaks from the wthLRH-1 samples. Genes up-regulated and down-regulated in response to TA was called by setting a probe-intensity threshold at the 95th percentile or 5th percentile, respectively, of array-wide probe

intensities. Motif searches were done by MEME-chip (*Machanick and Bailey, 2011*) and NR5A binding sequences were discovered by PROMO (*Farre et al., 2003*; *Messeguer et al., 2002*).

## AAV8 virus generation and retro-orbital infection protocol

Expression of 3X Flag-tagged WT or 2KR hLRH-1 in mouse liver was achieved using adeno-associated virus serotype-8 (AAV8) by cloning each respective cDNA into the viral gateway vector pAAV2.1-TBG (Penn Vector Core, Philadelphia, PA) downstream of the liver-specific promoter thyroxine-binding globulin (TBG). AAV8 virus expressing WT hLRH-1, 2KR or enhanced green fluorescent protein (eGFP) (AAV8 hLRH1, AAV8 hLRH1-2KR, or AAV8-eGFP) was amplified at the University of Pennsylvania Gene Therapy Vector Core. Tissue specificity and efficiency of infection was assessed in 8-week-old C57BL/6 male mice (The Jackson Laboratory, Bar Harbor, ME) infected via retro-orbital injection with AAV8-eGFP at concentration of $1 \times 10^{11}$ genome copies per ml (GC/ml), this concentration was used for subsequent experiments using AAV8-hLRH-1 and AAV8-2KR. Tissues were collected 14 days post-infection, as previously described (*Lu et al., 2012*), and analyzed for eGFP fluorescence by fluorescent microscopy or extracted for mRNA as described below. Mice were euthanized in accordance with the UCSF Institutional Animal Care and Use Committee under Ingraham lab protocol. Mice were perfused with PBS prior to collection of liver tissue for all subsequent biochemical and gene expression studies.

## Primary hepatocyte isolation

Primary hepatocytes were isolated from mice as previously described (*Silver et al., 2000*). Briefly, mice were anesthetized with Avertin (250 mg/kg) and perfused with pre-warmed perfusion buffer (HBSS supplemented with HEPES, pH 7.4) followed by perfusion with 75 ml digest buffer (HBSS/HEPES, pH 7.4, 0.3 mg/ml collagenase type 1, EDTA-free protease inhibitors). Digested liver was then dispersed on a 100 mm cell culture dish containing 25 ml cold Dulbecco's modified Eagle's medium (DMEM) (DMEM, 10% FBS, Penicillin/Streptomycin) and filtered through a 200-μm nylon membrane into a 50 ml Falcon tube. Hepatocytes were isolated by mixing filtrate with 24 ml Percoll solution (1x HBSS, 4 mM NaHCO3, pH 2.2) and centrifuged at 100 x g for 7 min at 4°C. Pellet containing hepatocytes was then washed with 40 ml DMEM and resuspended in desired media volume. Cells were plated on 6-well plates coated with collagen-1 and were allowed to attach overnight. The following morning DMEM was replaced with fresh DMEM supplemented with TA at the indicated concentration.

## Statistics

Data are represented as mean +/- SEM : *p<0.05; **p<0.005; ***p<0.001; ****p<0.0001. Statistical analyses were performed using Prism 5 (GraphPad) software. Statistical significance was determined by unpaired Student's T-test unless otherwise indicated.

## Acknowledgements

We wish to thank Dr S Hand for valuable reagents. We would also like to acknowledge Drs J Ward, M Asahino, B Shoichet, A Pierce, D Silver, and R Blind for experimental advice and discussion, as well as for critical reading of this manuscript. Funding sources that supported this work include an Innovation Grant from UCSF-PBBR/Roche grant, NIH P30 DK026743, R01-DK099722 and ADA 1-15-MI-08 to HAI, an NIDDK Supplemental Award R01-DK063592-563 S1 to HAI to support KAR, T32HD726330 and AHA 14POST2013-0048 to DAM, the QB3-564 Malaysia Program to support KKHA., and the Intramural Research Program at NIH, NIH 565 National Cancer Institute, Center for Cancer Research to support JSS.

## Additional information

### Funding

| Funder | Grant reference number | Author |
| --- | --- | --- |
| American Heart Association | 14POST2013-0048 | Diego A Miranda |

| National Institute of Child Health and Human Development | NIH T32 HD726330 | Diego A Miranda |
|---|---|---|
| National Institute of Diabetes and Digestive and Kidney Diseases | NIH R01 DK063592-S1 | Karmela A Ramos<br>Holly A Ingraham |
| QB3-Malaysia Program | | Kenny K-H Ang |
| National Cancer Institute | Intramural Research Program | John S Schneekloth Jr |
| University of California, San Francisco | UCSF-PBBR/Roche Innovation Grant | Holly A Ingraham |
| National Institute of Diabetes and Digestive and Kidney Diseases | NIH P30 DK026743 | Holly A Ingraham |
| National Institute of Diabetes and Digestive and Kidney Diseases | NIH R01 DK099722 | Holly A Ingraham |
| American Diabetes Association | 1-15-MI-08 | Holly A Ingraham |

The funders had no role in study design, data collection and interpretation, or the decision to submit the work for publication.

### Author contributions

MS, designed, optimized, and performed the primary screen; carried out in vitro, cellular and in vivo analyses, Conception and design, Acquisition of data, Analysis and interpretation of data, Drafting or revising the article; DAM, carried out in vitro, cellular and in vivo analyses, Conception and design, Acquisition of data, Analysis and interpretation of data, Drafting or revising the article; KAR, carried out in vitro, cellular and in vivo analyses, Acquisition of data, Analysis and interpretation of data, Drafting or revising the article; KK-HA, designed, optimized, and performed the primary screen; performed data analyses, Conception and design, Analysis and interpretation of data, Drafting or revising the article; EJF, designed, optimized, and performed the primary screen, Conception and design, Acquisition of data, Analysis and interpretation of data; CGW, AD, performed data analyses, Analysis and interpretation of data, Drafting or revising the article; LC, developed method for purification of recombinant full length hLRH-1 for all IVS assays, Acquisition of data, Contributed unpublished essential data or reagents; MRA, experimental design, Conception and design, Contributed unpublished essential data or reagents; YSK, carried out in vitro, cellular and in vivo analyses, Acquisition of data, Analysis and interpretation of data; RJF, developed method for purification of recombinant full length hLRH-1 for all IVS assays, Drafting or revising the article, Contributed unpublished essential data or reagents; JSS, experimental design, Conception and design, Acquisition of data, Drafting or revising the article; HAI, experimental design, supervised overall project and played a critical role in writing of manuscript and preparing figures, Conception and design, Analysis and interpretation of data, Drafting or revising the article

### Ethics

Animal experimentation: This study was performed in strict accordance with the recommendations in the Guide for the Care and Use of Laboratory Animals of the National Institutes of Health. All of the animals were handled according to approved institutional animal care and use committee (IACUC) protocols (#AN109147-01) of the University of California, San Francisco. Mice were euthanized in accordance with the UCSF Institutional Animal Care and Use Committee under Ingraham lab protocol.

## Additional files

### Supplementary files

• Supplementary file 1. List of forward and reverse qPCR primers used in our study.

• Supplementary file 2. Table providing details and work flow of the primary and follow-up secondary assays.

• Supplementary file 3. List of vendor for repurchasing candidate drug hits used in secondary filtering step and in immortalized cells, primary hepatocytes, and IVS assays.

• Source data 1. Z-scores from primary screen are listed for each drug tested (10 µM, 24 hr) after assaying for *APOC3* or *MUC1* transcripts in JEG3 cells stably expressing hLRH-1. Primary screening conditions and calculations for obtaining Z-scores are provided in the 'Material and methods' as well as in *Supplementary file 2*.

• Source data 2. List of genes obtained after profiling (Columns 1-3) and after ChIP-seq (Column 4) in HepG2 hLRH-1 cells; data are also represented in Venn diagrams in either *Figure 6* or *Figure 6— figure supplement 1*. Experimental conditions are summarized in each column header. All listed genes in profiling experiments (Columns 1-3) were changed up or down by 2-fold (log2 $\geq$1.0 or $\geq$-1.0) after normalization of data and statistical significance were determined by comparing datasets, as described in 'Materials and methods'.

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
