## [Decision Letter]

Thank you for sending your work entitled "A Gene-Expression Screen Identifies A Non-Toxic Sumoylation Inhibitor That Mimics SUMO-Less Human LRH-1 In Liver" for consideration at *eLife*. Your Tools and Resources article has been favorably evaluated by John Kuriyan (Senior Editor) and three reviewers, one of whom is a member of our Board of Reviewing Editors.

The reviewers were in agreement that the identification of a SUMOylation inhibitor that is of widespread utility would be a major achievement. The major questions raised during review were the ability of the new inhibitor to affect the SUMOylation and activity of other SUMOylated proteins, and its specificity for affecting SUMO-dependent gene expression.

Major conceptual issues to be addressed:

1) Endogenous gene expression.

Does tannic acid inhibit the SUMOylation of endogenous LRH-1 and modulate gene expression regulated by endogenous LRH-1?

2) Effects on other SUMO-dependent pathways.

More data is needed to address whether TA affects the modification and activity of other SUMOylated proteins. This is important to assess whether the inhibitor has widespread utility. The global effect of siUBC9 is clearly better than TA treatment (compare Figure 5) but this is the opposite way around for effects on LRH-1 SUMOylation. This suggests that TA might be a more specific LRH-1 regulator rather than a more generic SUMOylation inhibitor. How many and which other TA/SUMO-sensitive genes were identified in the HepG2 microarrays that are not regulated by LRH-1, but by other proteins/TFs?

3) Relative toxicity of TA.

TA seems less effective than ginkgolic acid (GA) in reducing LRH-1 SUMOylation (in particular mono-). The authors conclude that TA is an effective non-toxic SUMO inhibitor in cells. As an alternative explanation, could it be that TA is more tolerated by cells than GA because it is less effective in inhibiting SUMOylation of substrates required for cell viability? Assuming that an effective inhibitor would shut down most SUMOylation events in cells, a toxic effect is not surprising and is compatible with the lethality of SUMO1 ko mice. The authors should provide some more information about the properties of TA. At which concentrations and time points does TA become cytotoxic in cell lines versus primary hepatocytes (expand on Figure 3, longer time treatments than 5 hours)? Presumably TA as polyphenol cannot be delivered to mice in vivo; if so, this should be pointed out to the reader.

4) Specificity of TA for SUMO-dependent gene expression.

The authors could provide more information on the degree to which TA actually mimics the effects of LRH-1 deSUMOYlation. The numbers of genes presented in Figure 5 is very small and within this small group, the correlations between siUBC9 and TA treatment are essentially random. In the examples selected, they do appear to do the same thing but one would expect to see a high degree of concordance. The common effects of TA, siUBC9 and the LRH1 SUMO mutant should be better correlated across the various genome-wide datasets. Generally for the data in Figure 5, statistics are needed.

---

## [Author Response]

*Major conceptual issues to be addressed: 1) Endogenous gene expression. Does tannic acid inhibit the SUMOylation of endogenous LRH-1 and modulate gene expression regulated by endogenous LRH-1?*

We agree for a need to show that TA inhibits sumoylation of endogenous LRH-1, as was shown for Flag-hLRH-1 in multiple platforms (in vitro, in cells and in primary hepatocytes). However, despite repeated efforts, available anti-LRH-1 antibodies (commercial and non-commercial) fail to cleanly detect sumoylated and unmodified LRH-1 in either tissues or cell lines – this is best observed by the multiple, non-specific bands shown in the Western blot for Figure 7. They are suitable for detecting recombinant hLRH-1, as shown in Figure 4. We also find that immunoprecipitation of endogenous mLRH-1 or hLRH-1 is also quite inefficient (1-5%) when compared with pull-downs of tagged-hLRH-1 using anti-Flag.

To overcome this issue, we asked if TA would inhibit sumoylation of *endogenous* SF-1 using the human adrenal carcinoma H295R cell line – in this instance the Upstate anti-SF-1 antibody will detect sumoylated SF-1 in embryonic and adult tissues (Lee, Dev Cell. 2011). In H295R cells, we find that TA decreases *endogenous* sumoylated SF-1 (see new Figure 5). We also show that TA decreases endogenous sumoylated RanGap (new Figure 5—figure supplement 2).

As for endogenous genes, it should be noted that in primary hepatocytes treated with TA, we use uninfected hepatocytes that do not express hLRH-1 (Figure 7). Responses shown in this panel are those obtained after adding TA alone (no infection of AAV-hLRH-1). This has been clarified by labeling the X-axis with “Uninfected Hepatocytes + TA”.

*2) Effects on other SUMO-dependent pathways. More data is needed to address whether TA affects the modification and activity of other SUMOylated proteins. This is important to assess whether the inhibitor has widespread utility. The global effect of siUBC9 is clearly better than TA treatment (compare Figure 5) but this is the opposite way around for effects on LRH-1 SUMOylation. This suggests that TA might be a more specific LRH-1 regulator rather than a more generic SUMOylation inhibitor. How many and which other TA/SUMO-sensitive genes were identified in the HepG2 microarrays that are not regulated by LRH-1, but by other proteins/TFs?*

We provide several lines of evidence showing that TA will inhibit sumoylation of other proteins, both in vitro and in cells:

A) As shown previously, TA inhibits in vitro sumoylation of SF-1, IκBα and an androgen receptor peptide (Figure 4—figure supplement 2);

B) We repeated experiments to ask if TA will decrease general protein sumoylation in HepG2 cells (Figure 5). Levels of sumoylated proteins, as detected by anti-SUMO1 and anti-SUMO2, decrease following TA treatment. In fact, for specific sumoylated proteins, TA is actually more effective than siUBC9, as we report for both hLRH-1 and RanGap. For other proteins, siUBC9 appears to be more effective.;

C) As mentioned above, we were pleased to find that TA diminishes sumoylation of endogenous SF-1 in human adenocarcinoma H295R cells (Figure 5);

D) We also find that sumoylation of endogenous RanGap, which is considered the most robust sumoylated substrate, decreases in HEPG2 cells treated with TA (Figure 5—figure supplement 2);

E) Finally,in our screening assay, we used *MUC1* that is markedly upregulated by siUbc9 (30-fold by siUBC9), but is a poor LRH-1 target. We speculate that upregulation of *MUC1* by TA (10-fold) results from another sumoylated regulator being targeted by TA.

As for the comparison between TA and siUBC9, we found that LRH-1 (Figure 5 and Figure 5—figure supplement 1, Figure 5—figure supplement 2) and RanGap remain sumoylated in the face of a substantial knockdown of Ubc9. SiRNA-mediated knockdown over 72 h resulted in a 95% loss of *UBC9* transcripts, with a 60% decrease in UBC9 protein (Figure 5—figure supplement 1). This residual UBC9 protein likely accounts for the persistent levels of 1xSU-LRH-1 and SU-RanGap. Further, for both SF-1/LRH-1, UBC9 was one of the strongest interacting protein partners (Lee, Mol Cell Biol. 2005). Thus, it is likely that once UBC9 is charged with SUMO1/2, sumoylation of LRH-1 proceeds efficiently. Interestingly, targeting E2 by 2-D08 or by siUBC9 appears to be ineffective at eliminating sumoylation of LRH-1, whereas targeting E1 through TA is much more effective. We also found that depleting 1x-SU-LRH- in HepG2 and JEG3 cells is much more difficult than in primary hepatocytes – this may reflect inherent differences in the extent of sumoylation between immortalized and primary cells, as suggested by Ballail et al., Nat Comm 2014. Nonetheless, as correctly suggested by the reviewers, and as we discussed in the text, the efficacy of any given sumo-inhibitor is likely substrate-dependent.

*3) Relative toxicity of TA. TA seems less effective than ginkgolic acid (GA) in reducing LRH-1 SUMOylation (in particular mono-). The authors conclude that TA is an effective non-toxic SUMO inhibitor in cells. As an alternative explanation, could it be that TA is more tolerated by cells than GA because it is less effective in inhibiting SUMOylation of substrates required for cell viability? Assuming that an effective inhibitor would shut down most SUMOylation events in cells, a toxic effect is not surprising and is compatible with the lethality of SUMO1 ko mice. The authors should provide some more information about the properties of TA. At which concentrations and time points does TA become cytotoxic in cell lines versus primary hepatocytes (expand on Figure 3, longer time treatments than 5 hours)? Presumably TA as polyphenol cannot be delivered to mice in vivo; if so, this should be pointed out to the reader.*

To address this important issue, cellular toxicity of TA or GA was assayed in HepG2, JEG3 and primary cells following 24 h post-treatment. As expected GA was significantly more toxic than TA in JEG3 cells at both 5 and 24 h beginning at 15 µM (Figure 3 and new Figure 3—figure supplement 2), and in HepG2 cells at 24 h (Figure 5). By contrast, little toxicity is observed at the highest dose of TA (50-75 µM) when LRH-1 sumoylation is greatly diminished (Figure 5). Importantly, at lower non-toxic doses, GA (1 and 10 µM) is far less effective than TA at inhibiting LRH-1 sumoylation (new Figure 5). TA shows little toxicity in primary hepatocytes, even at higher doses. A similar trend was noted for GA when comparing HepG2 cells and primary hepatocytes (Ma et al., Oncotarget 2015) (Figure 7—figure supplement 1). TA appears more effective at inhibiting LRH-1 sumoylation in primary hepatocytes compared to HepG2 and JEG3 cells. In reference to the discussion below, this observation appears to uncouple the degree of toxicity and the extent of desumoylation.

It was suggested that the increased toxicity observed with GA might result from more effective shut-down of SUMOylation as compared to TA. However, we do not favor this hypothesis given our findings in primary hepatocytes (see above) and results in IVS assays showing that TA is a more effective inhibitor than GA (Figure 4). Moreover, it is debatable whether one can directly compare the embryonic lethal SUMO1 KO mouse model with GA toxicity and a shut-down of sumoylation. One could easily argue that the embryonic lethality in the developmental SUMO1 KO reflects altered functions of key proteins rather than general toxicity. Furthermore, for our purposes, the confounding toxic effects at doses of GA needed to inhibit sumoylation make it difficult to assess cellular readouts, such as gene expression.

*4) Specificity of TA for SUMO-dependent gene expression. The authors could provide more information on the degree to which TA actually mimics the effects of LRH-1 deSUMOYlation. The numbers of genes presented in Figure 5 is very small and within this small group, the correlations between siUbc9 and TA treatment are essentially random. In the examples selected, they do appear to do the same thing but one would expect to see a high degree of concordance. The common effects of TA, siUbc9 and the LRH1 SUMO mutant should be better correlated across the various genome-wide datasets. Generally for the data in Figure 5, statistics are needed.*

As noted here, it would be instructive to know how ChIP-Seq data sets overlap between TA + WT LRH-1 versus 2KR LRH-1. This information would be useful in determining how effectively TA mimics the transcriptional effects of LRH-1 desumoylation. This is an excellent point and one that we have earnestly tried to address. Unfortunately, we have failed repeatedly to maintain stable SUMOless LRH-1 cell lines that express equivalent protein levels as WT LRH-1. This same issue was encountered when we tried repeatedly to maintain stable SUMOless SF-1 cell lines in anything other than HEK293S cells.

Unlike immortalized cell lines, SUMOless SF-1 and SUMOless LRH-1 are stably expressed in tissues, as evidenced by our earlier study (Lee Dev Cell 2011) and this study, respectively. Obtaining ChIP-Seq and RNA-Seq datasets from primary hepatocytes expressing endogenous levels of both Flag-WT and SUMOless hLRH-1 with TA treatment will certainly be of interest, but begins to address a more specific biological question that is outside the scope of this current study. Understanding why SUMOless NR5As are stable in tissues but not in immortalized cell lines is also quite interesting, but again is beyond the scope of this study.

We have simplified Figure 6 (old Figure 5) and removed associated heat maps from supplemental figures, in part, because these data sets are not robust and limited by the use of microarrays rather than RNA-Seq. A subset of 42 genes was obtained after sorting by three parameters: 1) Genes upregulated by LRH-1; 2) LRH-1 targets by Chip-Seq and; 3) Genes up or downregulated by TA versus DMSO. While transcriptional effects of TA were observed for many of the 42 genes, we chose the top 3 target genes that contained a strong LRH-1 binding site (peak intensity) and yielded reproducible ChIP-qPCR data; both their expression and occupancy by LRH-1 were changed by TA. We have modified the text so as not to overstate these data. As requested, we have added the number of genes changed by siUbc9 versus TA (Figure 5 —figure supplement 3), which shows an overlapping subset of genes.